# Anti-Inflammatory Activity of Panduratin A against LPS-Induced Microglial Activation

**DOI:** 10.3390/biomedicines10102587

**Published:** 2022-10-15

**Authors:** Sopana Jamornwan, Tanida Chokpanuwat, Kwanchanok Uppakara, Sunhapas Soodvilai, Witchuda Saengsawang

**Affiliations:** 1Department of Physiology, Faculty of Science, Mahidol University, Bangkok 10400, Thailand; 2Chakri Naruebodindra Medical Institute, Faculty of Medicine, Ramathibodi Hospital, Mahidol University, Samut Prakan 10540, Thailand; 3Department of Basic Biomedical Sciences, Dr. William M. Scholl College of Podiatric Medicine, Rosalind Franklin University of Medicine and Science, North Chicago, IL 60064, USA

**Keywords:** neuroinflammation, microglial activation, panduratin A, *Boesenbergia rotunda*

## Abstract

Uncontrolled and excessive microglial activation is known to contribute to inflammation-mediated neurodegeneration. Therefore, reducing neurotoxic microglial activation may serve as a new approach to preventing neurodegeneration. Here, we investigated the anti-inflammatory effects of panduratin A against microglial activation induced by lipopolysaccharides (LPS) in the SIMA9 microglial cell line. We initially examined the anti-inflammatory properties of panduratin A by measuring LPS-induced nitric oxide (NO) production and the levels of pro-inflammatory cytokines (TNF-α, IL-1β, and IL-6). Panduratin A significantly reduced NO levels and pro-inflammatory cytokines’ production and secretion. In addition, panduratin A enhanced the production of anti-inflammatory cytokines IL-4 and IL-10. The anti-inflammatory effects of panduratin A are related to the suppression of the NF-κB signaling pathway. Together, these results demonstrate the anti-inflammatory properties of panduratin A against LPS-induced microglial activation, suggesting panduratin A has the potential to be further developed as a new agent for the prevention of neuroinflammation-associated neurodegenerative diseases.

## 1. Introduction

Neuroinflammation has been identified as a possible underlying cause for the development of neurological diseases, including Alzheimer’s disease (AD), Parkinson’s disease (PD), neuropsychiatric disorders, etc. [1,2,3,4,5]. Excessive microglial activation has been shown to play an important role in promoting neuroinflammation [6,7,8,9,10]. Chronic microglial activation promotes the production of pro-inflammatory mediators, such as nitric oxide (NO), cytokines, and chemokines, which contribute to damage to neurons and other cell types in the central nervous system [11,12,13,14]. These inflammatory factors can then drive further prolonged microglial activation, creating a positive feedback loop [15], and exacerbate neuroinflammation. Moreover, the elevated cytokines/chemokines can enhance the formation of amyloid-β (Aβ) aggregation and Tau hyperphosphorylation of fibrillary tangles, which are hallmark pathologies in AD [16]. Therefore, reducing chronic microglial activation is a potential anti-inflammatory therapeutic approach for preventing disease progression.

*Boesenbergia rotunda*, locally known as “Krachai”, is a member of the genus *Boesenbergia*, *Zingiberaceae* family. It is one of the traditional medicinal plants widely used as a food ingredient. Panduratin A shows several biological activities such as antioxidant, anti-apoptosis, antimicrobials, and anti-inflammatory [17,18,19,20,21,22,23]. In RAW264.7 macrophage cell models, panduratin A can inhibit the production of LPS-activated inflammatory mediators, including iNOS and COX-2 enzyme expression, NO, prostaglandin E_2_, and TNF-α through the suppression of NF-κB transcription activity [24,25]. Although several studies have reported the anti-inflammatory effects of panduratin A, its protective effects against neuroinflammation have not been investigated.

This study aimed to evaluate the anti-inflammatory effects of panduratin A isolated from *Boesenbergia rotunda* against LPS-induced microglial activation in a spontaneously immortalized microglial cell model (SIMA9). SIMA9 was isolated from the postnatal cerebral cortices of mice and has been previously used as a model for LPS-induced microglial activation [26,27,28,29,30,31]. Previous studies showed that SIMA9 cells expressed microglial marker proteins, CD68 and Iba-1, indicating that they exhibited a microglial property [26,32]. Lipopolysaccharides (LPS) were used as the inflammatory agent to promote prolonged neuroinflammation by enhancing microglial activation. LPS is a well-known inducer of inflammation through the interaction with Toll-like receptor 4 (TLR4), which is also activated by many endogenous ligands, including Aβ [33]. The effects of panduratin A on anti-inflammatory and pro-inflammatory cytokines’ expression and release, as well as the activation of NF-κB, were also investigated here.

## 2. Materials and Methods

### 2.1. Panduratin A Material

Panduratin A was isolated from the rhizomes of *Boesenbergia rotunda*. The extraction and isolation of panduratin A were conducted as previously reported [34]. Panduratin A was dissolved in DMSO and stored at −80 °C before use. The final concentration of DMSO was maintained at 0.01% in all experiments.

### 2.2. Cell Culture

SIMA9 cells were purchased from the American Type Culture Collection (ATCC^®^ CRL-3265^TM^, Manassas, VA, USA). SIMA9 cells were cultured in DMEM/F-12 (Gibco, Carlsbad, CA, USA), pH 7.4 supplemented with 10% (*v*/*v*) fetal bovine serum (Hyclone Laboratories Inc., Chicago, IL, USA), 5% (*v*/*v*) horse serum (Gibco, Carlsbad, CA, USA), and 1% (*v*/*v*) Penicillin-Streptomycin (Gibco, Carlsbad, CA, USA) at 37 °C under 5% CO_2_. The media were freshly replaced every other day. Cell cultures were passaged with a subculturing solution containing 1 mM EDTA, 1 mM EGTA, and 1 mg/mL glucose. Cells were allowed to adhere around 24 h before the experiment.

### 2.3. Cell Viability Assay

Cell viability was evaluated using 3-(4,5-Dimethyl-2-thiazolyl)-2,5-diphenyl-2H- tetrazolium bromide (MTT). To detect the cytotoxicity of panduratin A on SIMA9 cells, cells were plated in a 96-well plate at a density of 7.5 × 10^3^ cells/well on 0.01 mg/mL of Poly-D-lysine (MilliporeSigma, Burlington, MA, USA). After treating the cells with various concentrations of the compound, the media were gently removed, and the treated cells were incubated with the MTT solution (MilliporeSigma, Burlington, MA, USA) at a final concentration of 0.5 mg/mL for 3 h at 37 °C without being exposed to light. Next, the supernatant was carefully discarded, and the formed formazan crystals were dissolved using DMSO (MilliporeSigma, Burlington, MA, USA). The absorbance intensity was quantified using a microplate reader (Multiskan GO, Thermo Fisher Scientific, Waltham, MA, USA) at 570 nm.

### 2.4. Nitric Oxide Assay

The production of nitric oxide (NO) was measured using a Griess assay. Briefly, SIMA9 cells were cultured in a 96-well plate at a density of 7.5 × 10^3^ cells/well. The cells were then treated with the non-cytotoxic concentration of panduratin A for 24 h before being exposed to 10 ng/mL LPS for 24 h. The NO concentration was analyzed by measuring the nitrite released from LPS-treated microglia. For this, the supernatant from LPS-treated SIMA9 cells was collected to mix with Griess reagents (MilliporeSigma, Burlington, MA, USA), including 1% (*w*/*v*) Sulfanilamide in 5% (*v*/*v*) H_3_PO_4_ for 10 min and 0.1% (*w*/*v*) NED for 5 min at room temperature without being exposed to light. Then, the absorbance intensity was measured at 550 nm on a microplate reader (Multiskan GO, Thermo Fisher Scientific, Waltham, MA, USA). Sodium nitrite (NaNO_2_) (MilliporeSigma, Burlington, MA, USA) was used as the standard to make a standard curve for representative NO levels in each condition.

### 2.5. Reverse-Transcription Polymerase Chain Reaction

SIMA9 cells were plated at 2 × 10^5^ cells/well in 6-well culture plates. The cells were pre-treated with panduratin A for 24 h before being treated with 10 ng/mL of LPS for 6 h. Total RNA was isolated using a TRIzol^®^ reagent (Life Technologies Corporation, Singapore). Total RNA concentrations were then measured using a NanoDrop^TM^ 2000/2000c Spectrophotometer (Thermo Fisher Scientific, Waltham, MA, USA). The total RNA extraction was subsequently converted to cDNAs via an iScript^TM^ Reverse Transcription Supermix (Bio-Rad Laboratories, Hercules, CA, USA) following the manufacturer’s instructions. Next, expression levels of target genes from the cDNAs were generated by the RT-PCR detection system using an iTaq^TM^ Universal SYBR^®^ Green Supermix (Bio-Rad Laboratories, Hercules, CA, USA). The quantification cycle (Cq) value of a target gene was subtracted from the Cq values of the loading control GAPDH to represent mRNA expression levels.

### 2.6. Enzyme-Linked Immunosorbent Assay

A MILLIPLEX^®^ Mouse Cytokine/Chemokine Magnetic Bead Panel (MilliporeSigma, Burlington, MA, USA) was used to detect the levels of cytokines released from SIMA9 cells. For this assay, cells were plated into a 6-well plate at a density of 2 × 10^5^ cells/well and pre-treated with panduratin A for 24 h prior to exposure with LPS at 10 ng/mL for 24 h. After treatment, the culture media were collected, and cytokine release was quantitatively determined using a MILLIPLEX^®^ MAP Cytokine/Chemokine kit according to the manufacturer’s instructions. Briefly, the culture media were mixed with magnetic beads coated with antibodies in 96-black well plates. The plate was then incubated and shaken overnight at 4 °C. Next, biotinylated detection antibodies were added to each well and incubated for 1 h. Streptavidin-Phycoerythrin was then added and incubated for 30 min. After washing, Sheath Fluid was added and incubated for 5 min on a shaker. Finally, the fluorescence signals were measured using the Luminex MAGPIX^®^ system (MilliporeSigma, Burlington, MA, USA) to quantify cytokine concentrations.

### 2.7. Western Blot

SIMA9 cells were cultured at a density of 2 × 10^5^ cells/dish in a 60-mm dish and pre-treated with panduratin A for 24 h. The cells were then treated with 10 ng/mL LPS for 30 min to measure NF-κB expression and for 24 h to measure iNOS expression. After incubation, total protein lysates were extracted, and the total protein concentration was measured using a BCA assay kit (Thermo Fisher Scientific, Waltham, MA, USA). The proteins were separated by sodium dodecyl sulfate-polyacrylamide gel electrophoresis (SDS-PAGE) and transferred to Polyvinylidene difluoride (PVDF) membranes. After that, membranes were blocked with 5% (*w*/*v*) non-fat dry milk for 1 h, followed by incubation with primary antibodies, such as anti-iNOS (1:250, Abcam, Cambridge, UK), anti-phospho-NF-κB p65 Ser536 (1:1000, Santa Cruz Biotechnology Inc., Dallas, TX, USA), anti-NF-κB p65 (1:1000, Cell Signaling Technology, Danvers, MA, USA), and anti-GAPDH (1:10,000, Thermo Fisher Scientific, Waltham, MA, USA) at 4 °C overnight. After primary incubation, the membranes were rinsed 6 times with Tris-buffered saline with Tween-20 (TBS-T) for 10 min and incubated with secondary antibodies conjugated to horseradish peroxidase (HRP) (Jackson ImmunoResearch Laboratories Inc., West Grove, PA, USA) for 1 h. Membranes were then rinsed again, and chemiluminescence was detected using an enhanced chemiluminescence (ECL) reagent (MilliporeSigma, Burlington, MA, USA) for 5 min. The relative protein expression levels were normalized with GAPDH, and the quantification of target proteins was analyzed using ImageJ software (National Institutes of Health, Bethesda, MD, USA).

### 2.8. Statistical Analysis

The data were represented as the mean ± SEM from (at least) three independent experiments. The statistical differences among groups were analyzed using a one-way analysis of variance (ANOVA) followed by a Tukey’s multiple comparison using GraphPad Prism 9.4.1 software (GraphPad Software Inc., San Diego, CA, USA). *p*-values < 0.05 were considered statistically significant.

## 3. Results

### 3.1. Cytotoxic Potency of Panduratin A in SIMA9 Cells

First, to determine the non-toxic concentration of panduratin A (Figure 1A) on SIMA9 cells, cell viability was determined using an MTT assay. Cells were treated with panduratin A at different concentrations (0.1–50 µM) for 24 or 48 h. The IC_50_ of panduratin A on SIMA9 cells at 24 and 48 h were 48.96 ± 3.69 and 30.81 ± 2.61 µM, respectively (Figure 1B). At concentrations below 10 µM, panduratin A did not affect the viability of SIMA9 cells; therefore, these concentrations were chosen for further investigations.

### 3.2. Panduratin A Suppressed Nitric Oxide Production, iNOS Expression, and NF-κB Activation

Several studies have reported that elevated NO production is a possible etiology of several inflammation-related neurodegenerative disorders [35]. Excessive NO levels produced by chronically activated microglia drive the formation of reactive nitrogen species (RNS), leading to neuronal cell death [14]. In the present study, LPS was used as the inflammatory agent to promote microglial activation. Previous studies have reported that LPS can perform the neuroinflammation by enhancing the overwhelming inflammatory mediators such as cytokines/chemokines, NO, etc., produced by activated microglia [26,36]. To determine the effect of panduratin A on LPS-induced NO production in SIMA9 cells, we performed a Griess assay to measure NO levels. The cells were treated with panduratin A for 24 h before being exposed to LPS at 10 ng/mL for 24 h. As shown in Figure 2A, when stimulated with LPS, the NO levels increased by almost 3.33 ± 0.33-fold of the control. However, pre-treatment with panduratin A significantly and concentration-dependently reduced NO levels relative to LPS treatment alone. The treatment of N-Acetylcysteine (NAC) at 10 mM was used as the positive control in this study [37]. This inhibitory effect was not due to the cytotoxic effects of either LPS or LPS plus panduratin A since the MTT assay results suggest no changes in cell viability in any of these conditions (Figure 2B).

Inducible nitric oxide synthase (iNOS) is a key enzyme for the synthesis of NO from L-arginine [38]. To examine whether panduratin A reduced NO production through the suppression of iNOS expression, a Western blot analysis was used to detect iNOS protein expression levels. The treatment with LPS at 10 ng/mL for 24 h significantly increased iNOS expression higher than that of the control. However, this effect was blunted by pre-treatment with panduratin A for 24 h, in a concentration-dependent manner (Figure 2C). These results imply that panduratin A-driven anti-inflammatory effects against LPS-induced NO production might be, at least in part, mediated via the regulation of iNOS expression. Furthermore, we investigated the underlying mechanisms of the anti-inflammatory properties of panduratin A in LPS-stimulated SIMA9 cells. The NF-κB pathway is well-known to play a critical role in modulating the expression of inflammatory genes, such as iNOS, COX-2, TNF-α, IL-1β, IL-6, etc. [36,39]. Thus, we hypothesized that the anti-inflammatory effects of panduratin A may be related to the NF-κB signaling pathway. To test this hypothesis, levels of NF-κB-related proteins, such as p-NF-κB p65 Ser536 and NF-κB p65, were determined using a Western blot analysis. As shown in Figure 2D, the phosphorylation of NF-κB p65 was increased after 30 min of treatment with 10 ng/mL LPS. However, pre-treatment with panduratin A for 24 h concentration dependently reduced the levels of p-NF-κB p65, indicating it can suppress NF-κB activation.

### 3.3. Panduratin A Reduced TNF-α, IL-1β, and IL-6 Production in LPS-Induced Microglial Activation

Pro-inflammatory cytokines including TNF-α, IL-1β, and IL-6 are known to be regulated via the NF-κB-dependent signaling cascade [36,39]. These cytokines are released from activated microglia and drive damage in the surrounding brain tissues [11,12,13,14]. Therefore, in the present study, we investigated how panduratin A affects the production of pro-inflammatory cytokines in microglia treated with LPS. First, we investigated the effects of panduratin A on TNF-α, IL-1β, and IL-6 mRNA expression using RT-PCR. We found that the treatment of LPS at 10 ng/mL for 6 h increased all pro-inflammatory cytokine mRNA expression levels. Conversely, pre-treatment with panduratin A for 24 h inhibited the induction of cytokine expression by LPS stimulation in a concentration-dependent manner (Figure 3A,C,E). In addition to cytokine mRNA expression, cytokine release was also measured using an ELISA assay. The secretion of TNF-α, IL-1β, and IL-6 was concentration-dependently decreased in cells pre-treated with panduratin A for 24 h before being exposed to LPS at 10 ng/mL for 24 h (Figure 3B,D,F). These results suggest that the anti-inflammatory properties of panduratin A on LPS-induced microglial activation are involved in the suppression of pro-inflammatory cytokine mRNA expression and cytokine release.

### 3.4. Panduratin A Enhanced IL-4 and IL-10 Production in LPS-Induced Microglial Activation

Anti-inflammatory cytokines play an important role in neutralizing neuroinflammation in the AD brain. Several studies have reported that the production of anti-inflammatory factors is decreased in activated microglia, thereby promoting neuroinflammation and cognitive impairment in AD models [40,41]. To determine whether panduratin A could increase the production of anti-inflammatory cytokines in activated microglia, we first investigated the mRNA expression of the anti-inflammatory cytokines IL-4 and IL-10. The LPS treatment at 10 ng/mL for 6 h caused a significant reduction in IL-4 and IL-10 mRNA expression compared to untreated microglia. Conversely, panduratin A pre-treatment for 24 h, prior to exposure to LPS, increased the mRNA expression levels of both cytokines relative to LPS treatment alone (Figure 4A,C). In addition, the pre-treatment of panduratin A for 24 h significantly enhanced the IL-4 and IL-10 release from LPS-stimulated cells relative to LPS treatment alone for 24 h (Figure 4B,D). These data demonstrate that panduratin A promotes anti-inflammatory cytokine production and release.

## 4. Discussion

The etiology of AD remains elusive, and several studies have proposed that neuroinflammation is believed to be one of the causes for the development of progressive AD [1,2,3]. Microglial activation has been linked with pathophysiological processes of AD by increasing inflammatory mediators, resulting in damage to healthy neurons and impairing brain functions [11,12,13,14]. The present study illustrates the anti-neuroinflammatory activities of panduratin A and uncovers some of the underlying mechanisms of its action in activated microglial. In this study, LPS was used as an inflammatory agent to induce microglial activation based on evidence from previous studies indicating that LPS-induced neuroinflammation mimics several situations that occur in the AD brain [42,43,44]. Panduratin A efficiently suppressed microglial activation, resulting in decreased pro-inflammatory factors and increased anti-inflammatory mediators in LPS-stimulated SIMA9 microglial cells.

Panduratin A strongly decreased the mRNA expression levels and release of pro-inflammatory cytokines (TNF-α, IL-1β, and IL-6) and increased the releasing of the anti-inflammatory cytokines IL-4 and IL-10. Switching microglial phenotypes from the M2 anti-inflammatory stage to the M1 pro-inflammatory stage occurs in neuroinflammatory diseases [45,46]. The intra-hippocampal injection of Aβ_1–42_ was shown to activate M1 microglia by increasing pro-inflammatory cytokines, including IL-1β and IL-6 [41]. The Aβ_1–42_ injection also suppressed the M2 stage of microglia by decreasing the expression of the anti-inflammatory cytokine IL-10, which led to memory impairment [41]. In the present study, panduratin A decreased pro-inflammatory cytokine and increased anti-inflammatory cytokine production, which might indicate that panduratin A promotes microglial polarization from the M1 pro-inflammatory stage back to the M2 anti-inflammatory stage in LPS-induced SIMA9 cells. However, further studies are required to prove this hypothesis.

We also demonstrated that panduratin A decreased NF-κB, a key transcription factor that regulates the expression of pro-inflammatory factors. However, future studies are required to further investigate the mechanism of how panduratin A affects NF-κB activation. NF-κB inhibition might be a consequence of reducing the phosphorylation and degradation of IκBα, an upstream signaling factor of NF-κB [24]. Antioxidant action might also be an underlying mechanism of panduratin A since it has been shown to exert powerful antioxidant activities through the suppression of reactive oxygen species (ROS), and the generation and promotion of antioxidant-related enzymes, such as glutathione, catalase, and superoxide dismutase, in tert-Butyl hydroperoxide-injured HepG2 cells and in TNF-α-treated L6 cells [17,47]. A previous study has reported ROS act as central regulators of downstream inflammatory signaling, particularly with NF-κB activation in vascular endothelial cells and colon cancer cells [48,49,50]. It is possible that the anti-inflammatory effects of panduratin A might be related to its antioxidant property. A previous study in renal proximal tubular cells has shown that the co-treatment of panduratin A together with colistin reduced the effects of colistin on reactive oxygen species (ROS) production, mitochondrial damage, and apoptotic protein induction [51]. In another study using TNF-α to induce inflammation in rat skeletal muscle cells (L6 cells), the post-treatment of panduratin A inhibited ROS production by enhancing antioxidant-related enzymes [47]. Although in this study we demonstrated that panduratin A pre-treatment reduced the inflammatory response in microglia cells, future studies with the co-treatment and post-treatment of panduratin A in this model might provide more useful information on the mechanism underlying the anti-inflammatory effects of panduratin A.

## 5. Conclusions

Our data demonstrated, for the first time, that panduratin A isolated from *Boesenbergia rotunda* elicits neuroprotective effects against neuroinflammation by suppressing microglial activation. We found that panduratin A effectively inhibited the production of pro-inflammatory mediators, such as nitric oxide and pro-inflammatory cytokines, and it enhanced anti-inflammatory cytokine expression through the suppression of the NF-κB signaling pathway. Together, these results demonstrate panduratin A has the potential for preventing neurodegenerative diseases associated with neuroinflammation. Further studies in animal models are required to confirm the anti-inflammatory effects of panduratin A.

## Figures and Tables

**Figure 1 biomedicines-10-02587-f001:**
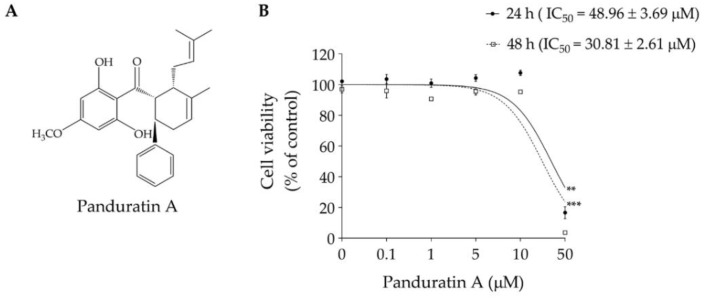
Cytotoxic potency of panduratin A in SIMA9 cells. (**A**) The chemical structure of panduratin A isolated from *Boesenbergia rotunda*; (**B**) viability of SIMA9 cells after exposure with various concentrations of panduratin A for 24 h (solid line) and 48 h (dashed line). Statistical analysis was performed using linear regression analysis to obtain the IC_50_ value of panduratin A in each time treatment. ** *p* < 0.01 and *** *p* < 0.001 compared to control (untreated cells).

**Figure 2 biomedicines-10-02587-f002:**
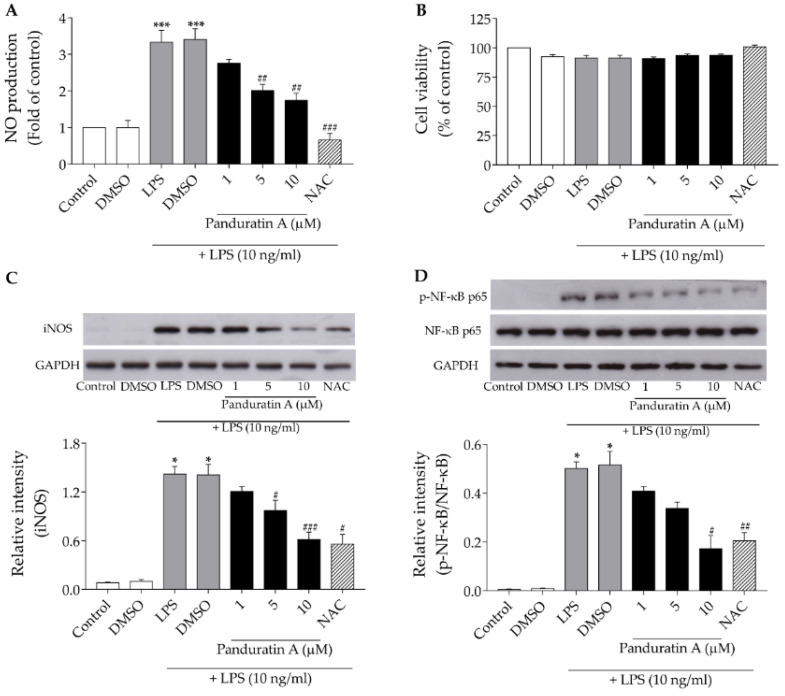
Effect of panduratin A against LPS-stimulated NO production, iNOS expression, and NF-κB activation. (**A**) Inhibition of LPS-induced NO production by panduratin A; (**B**) the viability of LPS-stimulated SIMA9 cells with or without panduratin A; (**C**) Western blot analysis of iNOS expression; (**D**) p-NF-κB p65 Ser536 and NF-κB p65 expression levels. NAC (10 mM) was used as the positive control. * *p* < 0.05 and *** *p* < 0.001 compared to control (untreated cells); # *p* < 0.05, ## *p* < 0.01, and ### *p* < 0.001 compared to LPS-treated cells (one-way ANOVA followed by Tukey’s test). Detailed information about the Western blotting can be found at Appendix A.

**Figure 3 biomedicines-10-02587-f003:**
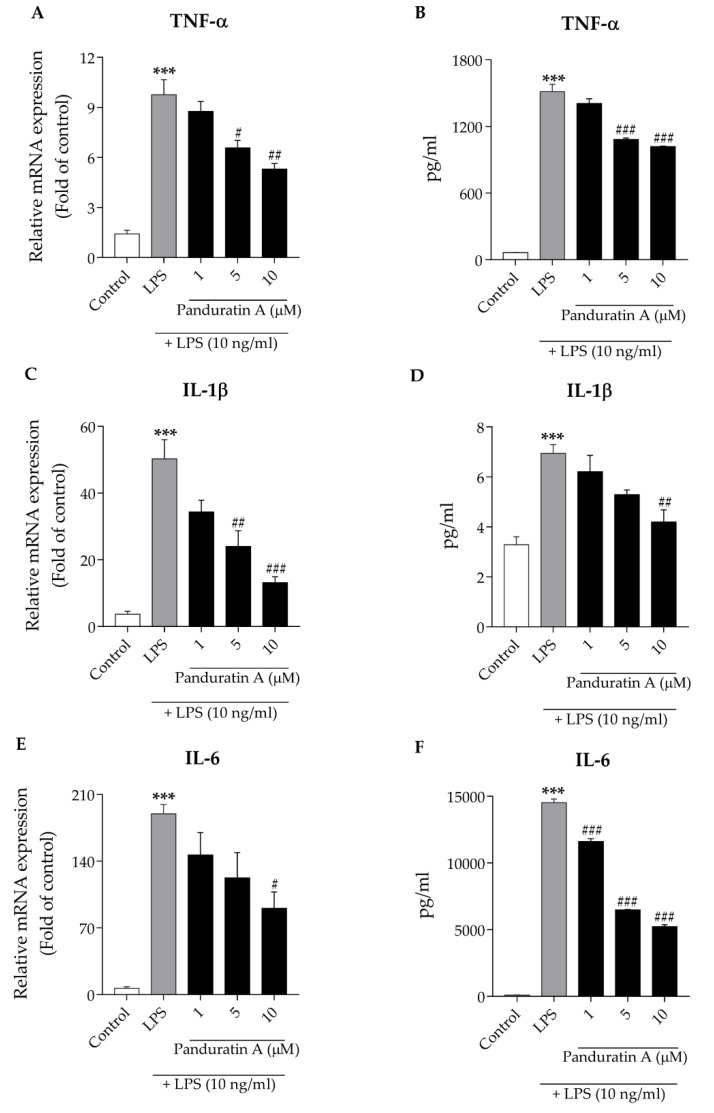
Effect of panduratin A on pro-inflammatory cytokine production against LPS stimulation. Panduratin A decreased mRNA expression of (**A**) TNF-α; (**C**) IL-1β; (**E**) IL-6. Panduratin A reduced cytokine release of (**B**) TNF-α; (**D**) IL-1β; (**F**) IL-6. The control was DMSO treatment. *** *p* < 0.001 compared to control; # *p* < 0.05, ## *p* < 0.01, and ### *p* < 0.001 compared to LPS-treated cells (one-way ANOVA followed by Tukey’s test).

**Figure 4 biomedicines-10-02587-f004:**
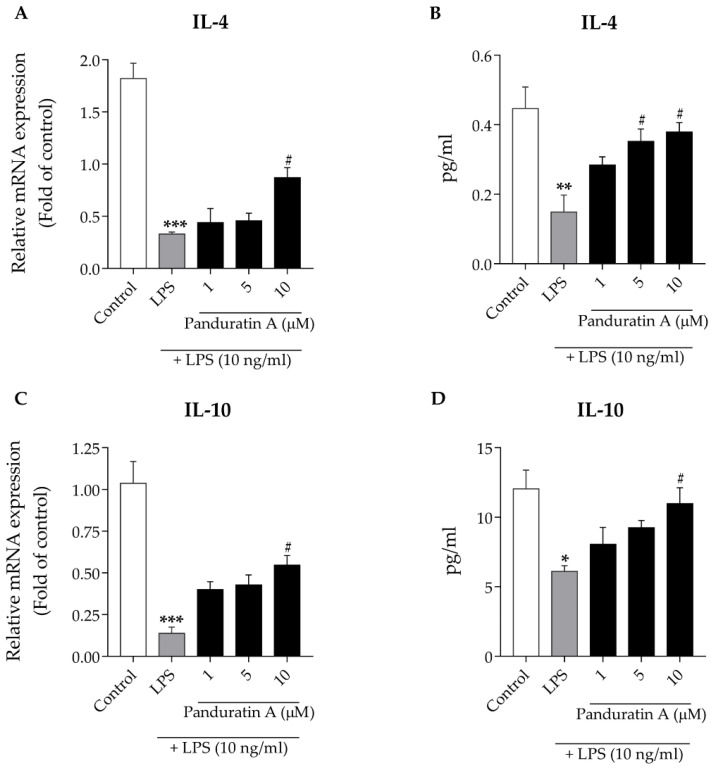
Effect of panduratin A on IL-4 and IL-10 production against LPS stimulation. Panduratin A increased mRNA expression of (**A**) IL-4; (**C**) IL-10. Panduratin A enhanced cytokine release of (**B**) IL-4; (**D**) IL-10. The control was DMSO treatment. * *p* < 0.05, ** *p* < 0.01, and *** *p* < 0.001 compared to control; # *p* < 0.05 compared to LPS-treated cells (one-way ANOVA followed by Tukey’s test).

## Data Availability

The data presented in this study are available on request from the corresponding author.

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
