# Peer review of "Anti-Inflammatory Activity of Panduratin A against LPS-Induced Microglial Activation"

_biomedicines, 2022, doi:10.3390/biomedicines10102587_

Round 1
Reviewer 1 Report
Minor comments
Introduction
1. Line 37 in page 1: after Aβ authors describe full name (amyloid-β)
Materials and Methods
1. Lines 60-63: In the beginning, the authors isolated panduratin A from rhizomes of Boesenbergia rotunda. In the next sentence, panduratin A was provided by Professor Patoomratana Tuchinda. It is not clear which one is the material of the experiment conducted.
Discussion
1. Lines 290-295 in page9: “Moreover, previous studies showed that panduratin A inactivated Akt and MAPK signaling (ERK, JNK, and p38) and blocked the formation of AP-1 complex by suppressing c-Jun and c-Fos expression, which in turn inhibited inflammatory responses in TNF-α-treated L6 cells, periodontitis, and allergy-related models [18,39,42]. Furthermore, panduratin A might also suppress inflammation by upregulating the PPARα/δ target gene in the oxazolone-induced atopic dermatitis-like model in hairless mice [43]. Another recent study showed that panduratin A can cross the blood-brain barrier, as panduratin A levels were measured in the rat brain after oral administration [44].” These paragraphs are not relevant to your study.
Major comments
The authors investigated the effect of panduratin A on LPS-treated microglia. The cells were pretreated with panduratin A for 24 h before being treated with 10 ng/ml of LPS for 6 h and performed NO assay, RT-PCR, ELISA, and Western blot. Did the authors try the co-treatment and post-treatment with LPS? Are any published data available? If available, describe the previous results with some references. If not available, I hope that the authors may describe some expected results in the Discussion section.
Reviewer 2 Report
The manuscript by Jamornwan S. et al. entitled 'Anti-inflammatory Activity of Panduratin A against LPS Induced Microglial Activation' investigates the ability of the natural compound Panduratin A to counteract LPS-mediated neuroinflammatory processes in a murine microglia cell line. Panduratin A is a molecule that has been well-studied over the years, and its pharmacological effects have been extensively investigated; however, no data are present in the literature about its effects on neuroinflammatory processes. In this context, the topic is of interest.
The data are well presented and argued, however, some aspects should be improved before publication.
- In the introduction and discussion parts, the authors reported literature data about the involvement of microglia in the progression of neurodegenerative diseases, speaking only of Alzheimer's disease. It is well known how altered microglia activities are involved in many central disorders, including psychiatric ones. Please add more information on this topic.
- In the results section, all figures are reported without specifying which type of statistic and post hoc test analyses have been performed. Please, provide information.
- Lane 176: please change “A previous study”. with “Several studies”. The involvement of NO production in neurodegenerative disorders is a well-known topic. Moreover, the authors reported a review as a bibliographic reference.
- Lane 261: rewrite the period as it is badly formulated
- As SIMA9 cells are not commonly used as a microglia cell line, the authors could provide some images, preferably stained with a microglial marker such as Iba1, to provide information on morphology after different treatments.
- Do the authors have an idea about a possible pharmacological target for Panduratin A? If it is already reported in the literature, please provide some information.
- Are the pg/mL of the cytokines analyzed by ELISA normalized to parameters related to cell confluence, such as the absorbance of mg protein? Please, report details.
Round 2
Reviewer 2 Report
The authors have improved the quality of the manuscript following my suggestions. Thus, I approved the publication of the reviewed paper in this journal.
Best Regards.